# Adaptive Surrogate Gradients for Sequential Reinforcement Learning in Spiking Neural Networks

**Korneel Van den Berghe**\*
Delft University of Technology

**Stein Stroobants**
Delft University of Technology

**Vijay Janapa Reddi**
Harvard University

**G.C.H.E. de Croon**
Delft University of Technology

## Abstract

Neuromorphic computing systems are set to revolutionize energy-constrained robotics by achieving orders-of-magnitude efficiency gains, while enabling native temporal processing. Spiking Neural Networks (SNNs) represent a promising algorithmic approach for these systems, yet their application to complex control tasks faces two critical challenges: (1) the non-differentiable nature of spiking neurons necessitates surrogate gradients with unclear optimization properties, and (2) the stateful dynamics of SNNs require training on sequences, which in reinforcement learning (RL) is hindered by limited sequence lengths during early training, preventing the network from bridging its warm-up period.

We address these challenges by systematically analyzing surrogate gradient slope settings, showing that shallower slopes increase gradient magnitude in deeper layers but reduce alignment with true gradients. In supervised learning, we find no clear preference for fixed or scheduled slopes. The effect is much more pronounced in RL settings, where shallower slopes or scheduled slopes lead to a $\times 2.1$ improvement in both training and final deployed performance. Next, we propose a novel training approach that leverages a privileged guiding policy to bootstrap the learning process, while still exploiting online environment interactions with the spiking policy. Combining our method with an adaptive slope schedule for a real-world drone position control task, we achieve an average return of 400 points, substantially outperforming prior techniques, including Behavioral Cloning and TD3BC, which achieve at most –200 points under the same conditions. This work advances both the theoretical understanding of surrogate gradient learning in SNNs and practical training methodologies for neuromorphic controllers demonstrated in real-world robotic systems.

## 1 Introduction

Spiking Neural Networks (SNNs) are a class of neuromorphic algorithms [1] that offer native temporal processing and significantly outperform conventional deep learning architectures in terms of energy efficiency across a wide range of applications [2, 3, 4]. However, training SNNs for control tasks remains challenging due to the non-differentiable nature of spiking neurons, which complicates gradient-based optimization. Surrogate gradients [5] are a popular solution to this issue. Gygax and Zenke [6] demonstrate that the gradient can severely deviate from its true value depending on the surrogate gradient chosen. Yet, how these deviations affect learning in deep or stacked architectures remains largely unexplored.

---

\*Corresponding email: korneel.vandenberghe@hotmail.be

39th Conference on Neural Information Processing Systems (NeurIPS 2025).

Moreover, leveraging the temporal processing capabilities of SNNs requires training on sequences rather than individual transitions. A critical component of this process is a warm-up period, a number of timesteps during which the hidden states of the network are stabilized before gradients are applied. Reinforcement Learning (RL) is a framework for sequential decision-making, which has been used to achieve superhuman performance on Atari games [7] and drone racing [8]. However, at its core, most RL algorithms assume the underlying process to be a Markov Decision Process (MDP), where the next state depends only on the current state and action. This assumption is often not satisfied in robotics [9, 10]. Frame-stacking acts as a work-around by explicitly adding a history of observations to approximate an MDP [11, 12, 13, 14]. This approach is inefficient and redundant for SNNs, which are inherently stateful and capable of encoding temporal dependencies internally.

A major challenge arises in robotic environments, such as drone control. Subpar initial policies often lead to early episode termination (e.g., due to crashing), preventing the collection of sufficiently long sequences to bridge the warm-up period. The agent cannot act long enough to gather data that would allow it to improve. Addressing this issue is key to unlocking the potential of sequence-based training with SNNs in real-world control tasks.

In this work, we analyze the effect of surrogate gradient settings on the gradient being propagated throughout the network and investigate the role of surrogate gradients across a spectrum of learning regimes, from supervised learning to online RL. We introduce a novel RL algorithm tailored for continuous control with Spiking Neural Networks, explicitly leveraging their inherent temporal dynamics without relying on frame stacking. As a case study, we demonstrate the efficacy of our approach by training a low level spiking neural controller for the Crazyflie quadrotor [15]. The controller is trained entirely in simulation and successfully transfers to the real-world platform, bridging the reality gap without the need for observation or action history augmentation. This work contributes both practical insights and theoretical implications for training temporal-aware, energy-efficient embodied agents.

## 2 Related Work

Several approaches have been proposed for combining Reinforcement Learning (RL) with Spiking Neural Networks (SNNs). Early work focused on biologically plausible learning rules, such as Spike-Timing-Dependent Plasticity (STDP), which have been successfully applied to tasks like source seeking [16] and maze navigation [17]. While these methods demonstrate the potential of SNNs for solving control tasks, they often lack the scalability and efficiency of modern deep RL techniques.

To benefit from the advances in deep RL, surrogate gradients can be used to enable end-to-end training of SNNs with standard RL algorithms. The Deep Spiking Q-Network (DSQN) algorithm [11] adapts the DQN algorithm [7] to the spiking domain. Although DSQN achieves robust performance and demonstrates significant energy savings on neuromorphic hardware [18], it is trained on single transitions and resets the network state at every step. Therefore, it resets the network state at every environment transition, preventing the model from capturing temporal dependencies across time. Moreover, as a value-based method, DSQN is inherently limited to discrete action spaces and is not well-suited for continuous control.

More generally, value-based RL algorithms that leverage recurrent architectures, such as R2D2 [19] rely on a warm-up period to stabilize hidden states before applying updates. This technique assumes the agent can gather long sequences of experience. However, in robotic control environments, such as drone flight, poorly initialized policies often result in early termination (e.g., due to crashing), preventing the network from bridging this period. This limits the practicality of recurrent value-based methods in high-risk real-world environments.

To address these limitations, policy-based approaches have been explored. SNN-PPO [20] applies the Proximal Policy Optimization algorithm to train SNNs in an on-policy manner using entire trajectory sequences. This method has been shown to successfully solve a variety of continuous control tasks from the MuJoCo suite [21]. However, on-policy methods like SNN-PPO can be challenging to use when sample cost is high. Furthermore, they do not propose a solution for bridging the warm-up period.

Off-policy approaches enable reusing past experience stored in the replay buffer. PopSAN [22] is an off-policy, actor-critic method designed for continuous control with SNNs. By leveraging a conventional ANN as the critic, PopSAN benefits from the stability of standard deep RL while using an energy efficient SNN-based actor. It achieves strong performance on several MuJoCo tasks in simulation and reports up to a 140x reduction in energy consumption compared to non-neuromorphic implementations [23]. Nonetheless, similar to DSQN, PopSAN requires the reset of the hidden states between transitions, limiting its ability to capture temporal dependencies across timesteps.

## 3 Methods

### 3.1 Online TD3BC with Jump-Starting Privileged Actor

To address the aforementioned challenges in sequential SNN training, we adapt the Jump-Start Reinforcement Learning (JSRL) framework [24]. It leverages a pre-trained guide policy to create a curriculum of starting conditions for a secondary policy. We implement a privileged, non-spiking actor, trained for a few epochs through TD3 [25], training is stopped when the policy can hover the drone for the warm-up period consistently. Its primary function is to bridge the critical warm-up period required by the SNN. Both the guiding policy and the non-spiking critic receive privileged information in the form of action histories.

A replay buffer is continuously populated throughout training with transitions generated by both the guiding actor (during the initial warm-up phase) and the spiking policy (during subsequent interactions). This hybrid approach bridges the early training period, where the spiking policy still gathers short sequences. Therefore, supervised learning from the guiding policy interactions is effective during early training.

The guide controller, which receives a privileged set of observations, including the action history, is used for the first $500 - N$ timesteps, after which the spiking policy interacts for the remaining $N$ steps, linearly increasing $N$ until the guide policy is only used for the warm-up period. Initially, the buffer is filled mostly with experience from the guiding policy. This introduces a risk of training a spiking policy that closely resembles the guide policy, which can be undesirable as the guide is usually not an expert policy. Therefore, the choice of the BC term weight is crucial to prevent the spiking policy from overfitting to the guide policy.

Once our spiking policy gains a baseline performance, and generates sufficiently long rollouts, we want to leverage the reward information, for which online RL is better suited. Inspired by TD3BC [26], we incorporate a Behavioral Cloning (BC) term into the RL objective function that utilizes the guide policy demonstrations stored in the replay buffer. The BC weight $\lambda$ decays exponentially, shifting from imitation to reward-based optimization. The BC term in our approach serves two purposes: avoiding large changes in policy behavior, improving training stability, and leveraging the demonstration data efficiently.

We call this approach TD3BC+JSRL, pseudocode can be found in subsection 8.4. The objective to be maximized can be defined as:

$$J_\pi = \mathbb{E}_{\tau \sim \mathcal{D}} \left[ \sum_{i=0}^{n_{\text{seq}}-1} \mathbf{1}_{i \geq n_{\text{warm-up}}} \left( Q_{\phi_1}\big(\mathbf{s}_{\tau,i}, \pi_\theta(\mathbf{s}_{\tau,i} \mid \mathbf{h}_{\tau,i})\big) - \lambda \big\| \pi_\theta(\mathbf{s}_{\tau,i} \mid \mathbf{h}_{\tau,i}) - \mathbf{a}_{\tau,i} \big\|^2 \right) \right]. \quad (1)$$

Where the history, $\mathbf{h}_{\tau,\mathbf{i}}$, is defined as:

$$\mathbf{h}_{\tau,i} := (\mathbf{s}_{\tau,0}, \mathbf{s}_{\tau,1}, \ldots, \mathbf{s}_{\tau,i-1}), \quad (2)$$
$$\mathbf{h}_{\tau,0} := \varnothing. \quad (3)$$

$Q_{\phi_1}$ in the first term represents the first critic network, as used in TD3. The variable $\tau$ denotes a sequence sampled from the replay buffer, with a length of $n_{seq} = 100$. Within this sequence, $\mathbf{s}_{\tau,i}$ and $\mathbf{a}_{\tau,i}$ correspond to the $i^{\text{th}}$ observation and action, sampled from the replay buffer, respectively. The hyperparameter $\lambda$ controls the strength of the BC regularization and decays exponentially over time; the BC coefficient $\lambda$ starts at $\lambda_0 = 0.2$ and decays exponentially ($\lambda \leftarrow 0.99\lambda$) after each epoch, shifting emphasis from imitation to reward optimization as training stabilizes. When the guiding

policy is of high quality, a larger $\lambda$ is preferred. Finally, $\mathbf{1}_{i \geq n_{\mathrm{warm-up}}}$ equals zero during the warm-up period ($n_{\mathrm{warm-up}}$), which in our case is 50 steps, corresponding to $0.5$ seconds in the drone control task. We zero-initialize hidden states, which causes only marginal performance differences [19]. The critic is not time variant and thus does not need a warm-up period.

## 3.2 Spiking Neural Networks for Continuous Control

Spiking Neural Networks leverage bio-inspired neuron models, which transmit information through binary spikes. The membrane potential is charged by an input current $I_{in}$, which incrementally charges its membrane potential $U$. Over time, the potential decays at a rate determined by the leak factor $\beta$. When the membrane potential exceeds a defined threshold $U_{thr}$, the neuron emits a spike, $s$, and its potential is subsequently reset. We employ a soft-reset spiking mechanism, which is described by:

$$U[t+1] = \beta U[t] + I_{in}[t+1] - s \cdot U_{thr}, \quad \text{where } s = \begin{cases} 1, & \text{if } U[t] > U_{thr} \\ 0, & \text{otherwise} \end{cases} \tag{4}$$

Hence, the spiking function $f(U[t])$ is the non-differentiable Heaviside function and calls for surrogate gradients [5] to enable gradient based optimization methods. During the backward pass, we act as if this Heaviside function was a parametrized sigmoid function, $\sigma(kx)$, which is the true when $k \to \infty$. For computational efficiency, the gradient of this parametrized sigmoid is approximated by the gradient of fast sigmoid ($fs(k \cdot x) = \frac{k \cdot x}{1 + k \cdot |x|}$) [27]. To avoid exploding gradients, the actual gradient of the fast sigmoid is normalized by $k$, which leaves us with the derivative in Equation 5;

$$\frac{d}{dx} fs(k \cdot x) \cdot \frac{1}{k} = \frac{k}{(1 + k \cdot |x|)^2} \cdot \frac{1}{k} = \frac{1}{(1 + k \cdot |x|)^2}, \tag{5}$$

where a larger $k$ relates to a steeper slope of the parametrized sigmoid, reducing the range of inputs for which a gradient exists.

SNNs work in a spike-based domain. To encode continuous values to spikes and vice versa, population based encoding and decoding are used. Specifically, the first linear layer encodes continuous input values into currents, and a final layer decodes the output spikes into motor commands. This approach has been proven successful in previous work [28].

Our controller is a feedforward SNN with input size 18, two hidden layers and 4 output neurons. The two hidden layers of the network are of size 256 and 128 respectively and make use of Leaky Integrate-and-Fire neurons. This architecture enables analysis of surrogate gradient effects across multiple layers, sufficient capacity for continuous control, and is small enough for deployment on the resource constrained Crazyflie.

## 3.3 Surrogate Gradients in Deep Networks

Training SNNs via backpropagation requires the use of surrogate gradients to approximate the derivative of the non-differentiable spike function. A critical hyperparameter in this process is the slope $k$ of the surrogate function, which determines the sensitivity of the gradient near the spiking threshold.

We adopt the gradient of a fast sigmoid function, see Equation 5, and examine slope configurations ranging from shallow ($k = 1$) to steep ($k = 100$). We analyze a 4 layer SNN, with all layers having 64 neurons, layer 0 corresponds to the input layer, while layer 4 relates to the output. The results on the plots are the average of running the test 100 times. As shown in Figure 1a, steeper slopes closely resemble the Dirac delta function but restrict non-zero gradients to a narrow input range. In contrast, shallower slopes lead to a broader range of non-zero gradients. This increases the average gradient magnitude, particularly in deeper layers (Figure 1b). Note that the average gradient magnitude is influenced by the proportion of neurons with zero gradients in each layer.

While the surrogate gradient's slope affects the quantity of weight updates per backward pass, it also introduces noise into the gradient computation [6]. Since the true gradient for deeper network layers does not exist, we analyze the relationship between a steep surrogate gradient ($k = 100$), which

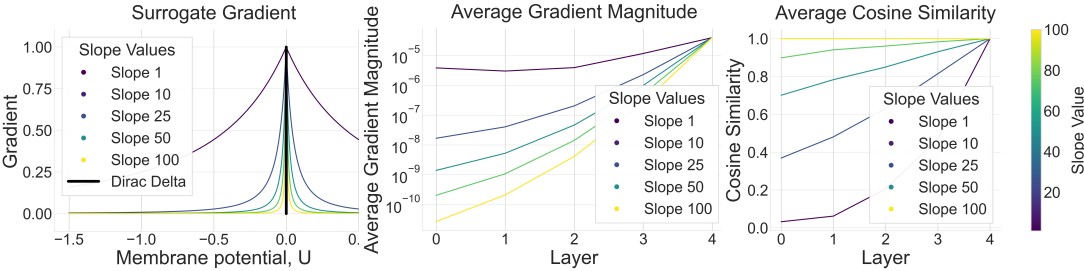

**(a)** The slope of the surrogate gradient dictates the range of inputs for which a gradient exists.

**(b)** A more shallow slope carries the gradient deeper through the network, suffering less from vanishing gradients.

**(c)** A shallower slope introduces bias and variance, computed using the cosine similarity.

**Figure 1:** Surrogate gradient slope-magnitude-alignment tradeoff in deep SNNs. (1a) Shallow slopes ($k$=1) provide gradients over wider input range than steep slopes ($k$=100). (1b) This prevents vanishing gradients in deeper layers, critical for multi-layer SNN training. (1c) However, shallow slopes introduce directional noise (cosine similarity $\rightarrow$ 0), which aids RL exploration but hinders supervised convergence. Network: 4 layers of 64 neurons; layer 0 = first hidden, layer 4 = output.

approximates the true gradient $\nabla W_i$ more truthfully than a shallow slope, and shallow surrogate gradients, denoted as $\tilde{\nabla} W_i$, using cosine similarity:

$$\text{cosine similarity} = \frac{\nabla \mathbf{W}_i \cdot \tilde{\nabla} \mathbf{W_i}}{\|\nabla \mathbf{W}_i\| \|\tilde{\nabla} \mathbf{W_i}\|}. \tag{6}$$

Figure 1c shows that the cosine similarity for shallow slopes reduce to 0, therefore, the weight updates in deeper networks become essentially random.

We propose an adaptive surrogate gradient scheduling approach. Shallow slopes, which introduce stochasticity into the gradient computation, can facilitate the update of a broader set of connections in deeper networks, enhancing exploration in RL, and increase the gradient magnitude. Conversely, steep slopes yield more precise gradient estimates, which are advantageous once the agent's performance stabilizes and fine-tuned optimization is desired. This mechanism provides a natural means of balancing exploration and exploitation throughout training.

Three schedules are analyzed:

- *Fixed*: The surrogate gradient slope is held fixed throughout training.
- *Interval*: The surrogate gradient slope is made steeper according to a fixed time schedule.
- *Adaptive*: The surrogate gradient slope is adapted as function of the achieved reward.

The adaptive slope scheduler tracks both the reward value and its derivative (see Equation 7), allowing the slope to increase when performance improves and stabilize once it saturates. This keeps the gradient noisy when exploration is needed and sharp when fine-tuning. Note that the term depending on $r_{t-i}$ is largely responsible for maintaining the slope when maximum performance is reached, and thus prevents destroying progress.

$$k_t = \frac{1}{10} \sum_{i=0}^{9} \left[ 0.5 r_{t-i} + 0.5 r'_{t-i} \right], \tag{7}$$

where $k_t$ is the slope at time $t$, $r_{t-i}$ is the reward score at time $t-i$ and $r'_{t-i}$ is the first order derivative of the reward score at time $t-i$. Furthermore, $k$ is clamped between 1 and 100, the same limits as used in Figure 3.

### 3.4 Asymmetric Actor-Critic

While the actor is spiking, the critic is a feedforward ANN, which receives all observed states and an action history of 32 timesteps. This asymmetric setup has been demonstrated to successfully leverage

the improved training stability of ANN [22, 23]. The jump-starting actor used in this work, is also an ANN network with the same privileged inputs as the critic. This critic is only used to stabilize training, and is not deployed on the drone at inference time.

# 4 Experimental Setup

## 4.1 Quadrotor Position Control Task

We validate our approach on low-level quadrotor control, a challenging domain where (1) early crashes prevent long sequences needed for SNN warm-up, directly testing our TD3BC+JSRL solution, and (2) the temporal dynamics stress-test surrogate gradient choices across extended rollouts. he Crazyflie 2.1 platform [15] provides 18-dimensional state vectors of position, velocity, orientation, and angular velocity. The controller outputs motor commands $a_t \in \mathbb{R}^4$ at $100Hz$, matching the onboard sensor update rate.

Figure 2 illustrates the information flow between the Crazyflie platform and the spiking controller.

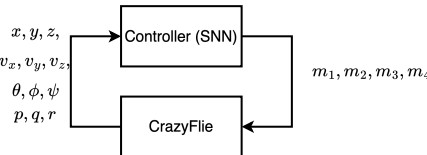

**Figure 2:** The spiking controller receives $(x, y, z)$, linear velocity $(v_x, v_y, v_z)$, orientation angles $(\theta, \phi, \psi)$, and angular velocities $(p, q, r)$ inputs and outputs motor commands $(m_1, m_2, m_3, m_4)$.

Episodes terminate after $500$ timesteps ($5$ seconds) or upon crashes. This limit creates the core challenge our method addresses: early training policies crash within 100 steps, preventing data collection beyond the 50-step warm-up period required for stable SNN gradients.

Training completes in 6 hours on an M4 Pro MacBook (24GB RAM). We compare four approaches (Table 1):

**Table 1:** Method comparison. Only TD3BC+JSRL (ours) combines online learning with warm-up bridging and demonstration leveraging, enabling sequence-based SNN training despite early crashes.

| Method | Type | Leverages Reward | Warm-up | Uses Demos |
|---|---|---|---|---|
| BC | Offline | X | X | ✓ |
| TD3BC | Offline | ✓ | X | ✓ |
| TD3 | Online | ✓ | X | X |
| TD3BC+JSRL (Ours) | Online | ✓ | ✓ | ✓ |

## 4.2 Simulation Environment and Reward Structure

The controller is trained in simulation, relying on a dynamics model which has demonstrated sim-to-real bridging capabilities [14].

We employ a linear curriculum learning approach where the reward structure increases difficulty as training advances. The total reward at each timestep is computed using Equation 8.

$$r_t = C_{rs} - C_{rp}\|p_t - p_{\text{des}}\|^2 - C_{rv}\|v_t - v_{\text{des}}\|^2 - C_{rq}\|q_t - q_{\text{des}}\|^2 - C_{ra}\|a_t - C_{rab}\|^2, \quad (8)$$

where $C_{rs}$ is a survival bonus encouraging long episodes, $C_{rp}, C_{rv}, C_{rq}$ penalize deviations from desired states, such as $p_{\text{des}}$, and $C_{ra}\|a_t - C_{rab}\|^2$ penalizes deviations from hover throttle $C_{rab}$. We apply curriculum learning by linearly interpolating penalty coefficients from lenient to strict values over training. The reward is explained in detail in subsection 8.2.

# 5 Results

## 5.1 Surrogate Gradients During Training

We now analyze the effect of surrogate gradient slope choices on both fully supervised, Behavioral Cloning (BC), and fully online training algorithms, TD3 [25]. To eliminate the effect of the warm-up during training for this analysis, we do the following. We stack a history of observations, process multiple forward passes per observation and reset the SNN in between subsequent actions, similar to previous RL implementations for SNNs. The dataset used for BC has been gathered with the privileged actor introduced in subsection 3.1. The following experiments do not use a reward curriculum, nor learn from the temporal dimension.

While slope choice barely affects supervised learning, online RL strongly prefers shallower slopes whose induced gradient noise enhances exploration, similar to parameter noise [29], albeit with higher variability across runs. Poor intermediate updates can corrupt the replay buffer with low-quality experiences, hindering convergence. Shallower slopes introduce noise in the policy update, and therefore heighten this risk, making training less stable.

We observe that scheduled slope settings firstly reduces the number of epochs to reach a reward of 100 by a factor $\times 4.5$ compared to steeper slopes. Secondly, final performance of the trained agents lie in the same regime as fixed slope experiments.

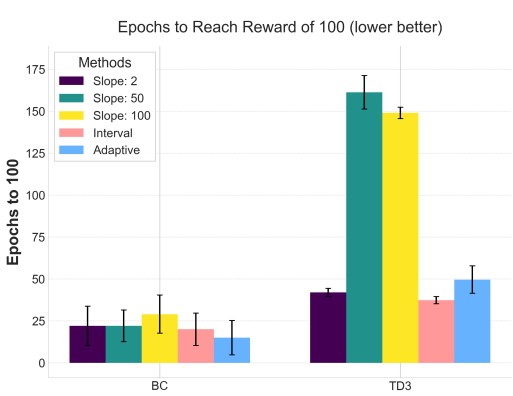

**(a)** In BC, convergence speed is largely unaffected by the surrogate slope, as the noisier updates from shallow slopes are compensated by broader gradient propagation. In contrast, RL benefits significantly from shallower slopes, which enhance exploration through gradient noise. Scheduled slopes achieve competitive convergence speeds without manual tuning.

**(b)** For BC, final performance remains largely invariant to the slope setting, confirming the robustness of supervised training. In RL, however, shallow slopes yield superior asymptotic rewards due to their exploration-promoting noise. Scheduled slopes, specifically adaptive scheduling performs competitively, approaching the performance of optimally tuned shallow slopes while improving training stability.

**Figure 3:** Comparison between fixed and scheduled surrogate gradient slopes for both BC and RL settings. While BC performance is largely slope-invariant, RL benefits from shallower or scheduled slopes, which improve exploration and training efficiency. Scheduling effectively balances stability and performance, reducing the need for extensive slope tuning.

While using an interval or adaptive schedule for the surrogate gradient slope does not always result in the best performing network, we find that it can stabilize training. Moreover, it achieves a near-optimal performance on both training speed and final performance. As a result, such schedules eliminate the need for exhaustive hyperparameter sweeps across different slope settings to identify the best performing ones.

## 5.2 Training on Sequences with Reward Curriculum

To fully leverage the temporal processing capabilities of SNNs in RL, we extend training from single transitions to full sequences (subsection 3.1) and introduce a reward curriculum that gradually increases penalties on position, velocity, and action magnitude to encourage stable, robust behavior.

All experiments use an adaptive slope scheduler for surrogate gradient slope adjustment. We compare BC, TD3BC, TD3, and TD3BC+JSRL. BC and TD3BC are offline RL algorithms trained on dataset sequences, gathered with the guiding policy used in our TD3BC+JSRL setup, while TD3 and TD3BC+JSRL are online algorithms sampling sequences from the environment. Unlike TD3, which starts from scratch, TD3BC+JSRL benefits from a privileged guiding policy during the first $n$ steps of rollouts. All approaches are evaluated under the same reward curriculum.

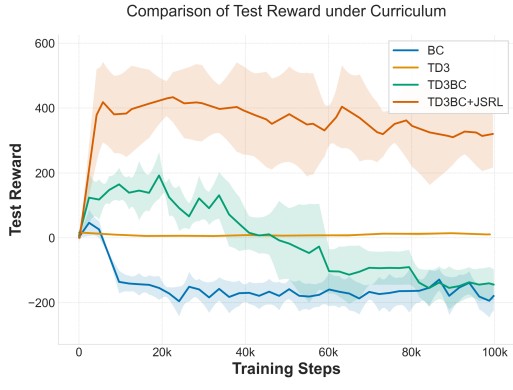

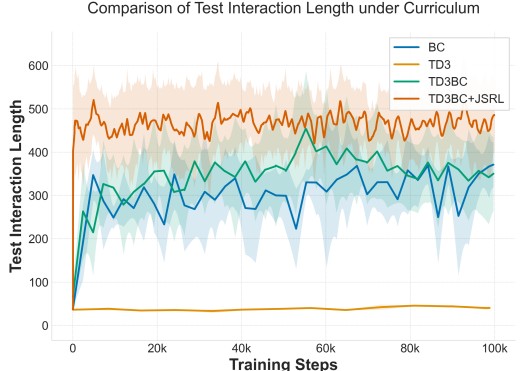

**(a)** Offline methods (BC and TD3BC) struggle as the reward function adapts, failing to generalize beyond the initial dataset, due to the lack of online interactions. TD3BC+JSRL is able to adapt to the curriculum throughout training.

**(b)** While all methods, except TD3, eventually learn to fly, TD3-trained policies frequently terminate early due to unstable exploration. TD3BC+JSRL achieves longer and more stable flight trajectories.

**Figure 4:** Comparison of training approaches under a reward curriculum. All experiments were run with 5 different seeds. As the curriculum increases difficulty of the task, we find that only TD3BC+JSRL is able to adapt to the changing curriculum. During testing, we average the results from 20 runs.

TD3BC+JSRL achieves 400-point average reward under curriculum learning, while BC, TD3BC, and vanilla TD3 fail to exceed $-200$ points (Figure 4a). This 600-point gap validates both components of our approach: leveraging demonstrations (vs. TD3's from-scratch learning) and online adaptation (vs. BC/TD3BC's static datasets).

As the curriculum tightens, BC and TD3BC, trained on static datasets containing rollouts of the guiding actor, fail to generalize, as shown in Figure 4a. While TD3 has access to evolving interactions, it struggles to gather meaningful sequences early in training, often resulting in premature episode terminations, as seen on Figure 4b, failing to bridge the warm-up period. Our proposed TD3BC+JSRL approach, which leverages new rollouts guided by the policy, demonstrates robust learning even under a challenging curriculum. The inclusion of online rollouts enables the policy to adapt to the evolving reward landscape, leading to substantial performance improvements.

The hyperparameters, curriculum and training details used for the experiment are part of the supplementary materials, subsection 8.2.

### 5.3 Bridging the Reality Gap

We quantitatively evaluated the computational efficiency of our sequential SNN approach using NeuroBench [2]. The trained SNN is compared to a feedforward ANN that achieves similar performance, trained using TD3. This controller requires explicit action history to control the drone successfully. Results show that temporally-trained SNNs match ANN performance while exhibiting distinct computational traits. Despite a higher memory footprint, SNNs benefit from activation sparsity and primarily use energy-efficient accumulates (ACs) instead of energy-hungry multiply-accumulates (MACs), making them well-suited for neuromorphic deployment. While the number of dense operations of the SNN is much higher than the ANN to which we compare, the nature of the underlying ACs opens the opportunity for energy efficient compute. Using the methodology proposed by Davies *et al.* [30], we can compute a rough energy consumption estimate of $9.7 \times 10^{-5} mJ$, see subsection 8.3 for further details.

**Table 2:** NeuroBench performance comparison between ANN (64-64 architecture) and temporally-trained SNN (256-128 architecture) using TD3BC+JSRL. While dense operations are computed ignoring activation and connection sparsity, effective operations reflect the sparsity aware number of operations, as defined in NeuroBench [2].

| Model | Reward | Footprint (kb) | Activation Sparsity | SynOps Dense | SynOps Eff_MACs | SynOps Eff_ACs | Requires History |
|-------|--------|-----------------|---------------------|---------------|------------------|-----------------|------------------|
| *ANN* | 447 | 55.3 | 0.0 | $13.7 \times 10^3$ | $13.7 \times 10^3$ | 0.0 | True |
| *SNN* | 446 | 158.3 | 0.79 | $37.9 \times 10^3$ | $4.6 \times 10^3$ | $12.2 \times 10^3$ | False |

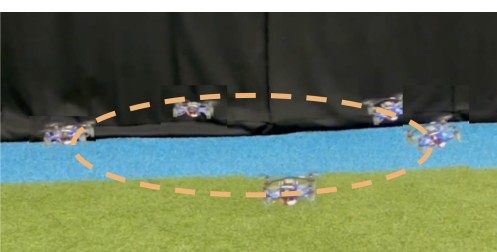

**Figure 5:** When deployed on the Crazyflie, the spiking actor exhibits oscillatory behavior but always successfully performs maneuvers such as circular flight.

When deployed on the Crazyflie, the SNN controller exhibits slightly oscillatory behavior. However, the SNN is still able to execute complex maneuvers like circles (see Figure 5), figure-eight and square trajectories without failure. ANN controllers, benefiting from full action history, show smoother control. However, when comparing to our SNN to an ANN which does not receive action history, we find that the ANN can no longer control the drone, as shown in Table 3.

To improve stability, future work could incorporate angular velocity penalties in the reward function, use throttle deviation outputs rather than absolute throttle settings, train for longer, or increased control frequency, as prior work has shown smoother SNN control at higher rates [14, 28].

Compared to ANN controllers from Eschmann et al. [14], SNNs trained with our approach achieved lower position error under ideal conditions but had reduced reliability. Notably, while ANNs without action history failed to control the drone. A demonstration video is available online[2].

**Table 3:** Comparison of neural network models for position control and trajectory tracking tasks. The models include an ANN with action history, an ANN without action history and an SNN trained with TD3BC+JSRL without action history. The mean position error of the deployed SNNs is benchmarked against the best-performing ANN policy [14]. Position error is measured as the average xy-plane error (in meters), and trajectory tracking error is evaluated as the average error across figure-eight and square-following tasks. The ANN without action history did not manage to stabilize the quadrotor.

|  | ANN action history [14] | ANN no action history [14] | SNN no action history [Ours] |
|--|--------------------------|----------------------------|------------------------------|
| Position Error [m] | 0.1 | 0.25 | 0.04 |
| Trajectory Error [m] | 0.21 | NA | 0.24 |

### 5.4 Ablation Study

To analyze the contribution of each component of the algorithm, an ablation study is performed. We first remove only the BC-term, then only the jump-start period. Lastly, we analyze the result with no BC-term and no Jump-Start.

Removing the BC term allows the SNN controller to eventually show improvements in performance, but convergence becomes substantially slower, approximately 15 times more training steps are required to achieve a reward of 100. In contrast, eliminating the jump-starting actor entirely prevents

---

[2]https://www.youtube.com/playlist?list=PLAS2CClQ48jX2R-tqza9kRPx0NW1EfKj0

the SNN from collecting sequences long enough to bridge the warm-up period, resulting in the network failing to learn stable flight altogether. As expected, removing both terms, leads to similar behavior. We find the BC term to improve training efficiency when the jump-start period allows bridging the warm-up period, and fills the buffer with several demonstrations from the guiding policy.

**Table 4:** Ablation study analyzing the impact of the BC-term and Jump-Start period on training performance. Reported values show the average final reward and the number of environment steps required to reach a reward of 100.

| BC-term | Jump-start | Reward | Steps to 100 |
|---------|-----------|--------|--------------|
| Yes | Yes | $412 \pm 6.72$ | $2200 \pm 124$ |
| No | Yes | $334 \pm 25.59$ | $33030 \pm 1600$ |
| Yes | No | $32 \pm 1.2$ | NA |
| No | No | $36 \pm 4.1$ | NA |

## 6    Conclusion

This work addresses key challenges in training Spiking Neural Networks (SNNs) for Reinforcement Learning (RL) by combining theoretical insights into surrogate gradient slope settings with a novel RL approach for sequential training. We demonstrate that the surrogate gradient slope plays a critical role in optimization dynamics, affecting both gradient alignment and the gradient magnitude. We find that shallower slopes increase the gradient magnitude at the cost of alignment with the true gradient. In supervised learning approaches, these two effects balance out. However, in RL approaches, we find a strong preference towards shallower slopes. We propose adaptive slope scheduling strategies that improve training stability and performance. We demonstrate that adaptive slope scheduling improves training efficiency by $\times 4.5$ compared to a fixed slope of 100.

To address the limitations of standard RL algorithms in temporally extended tasks, we introduce a jump-start framework that leverages privileged policies to bootstrap training. This enables effective sequence-based learning in unstable control tasks, such as drone flight, where subpar controllers fail to generate experience long enough to bridge the warm-up period, which is necessary to efficiently train stateful networks. We find our method to enable SNNs to achieve a final average reward of 400 points, while other methods fail to surpass a final performance of $-200$ points. We demonstrate the performance of our SNN receiving only the latest state information to be competitive to an ANN which receives explicit action history. A comparable ANN without this action history fails to control the Crazyflie successfully.

Our method depends on the availability of a guiding policy. Although this policy can be any function, even one leveraging privileged observations and does not need to be deployable at inference, it must still produce stable behavior early in training, which might not always be trivial to obtain.

## Code Availability

The implementation and experiment scripts used in this work are publicly available at github.com/korneelf1/SpikingCrazyflie. The repository contains detailed instructions to reproduce all reported results.

## 7    Acknowledgments

Author Guido de Croon gratefully acknowledges support from the Dutch Research Council (NWO) under grant number 20663 of the VICI personal grant programme.

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

# 8 Supplementary Materials

## 8.1 Effect of Surrogate Gradients on Gradient Alignment

When training spiking neural networks (SNNs), the non-differentiability of the spiking function is often handled by replacing it with a smooth surrogate function. In this section, we formalize how the slope of the surrogate gradient affects weight updates in deeper networks, supporting the main paper's findings on cosine similarity decay. We replace the binary spike with a sigmoid to obtain a tractable closed-form baseline, however, we believe the qualitative findings hold for spiking neurons as well.

**Network Setup.** We consider an ANN feedforward network with two hidden layers and sigmoid activations. Let $\mathbf{x} \in \mathbb{R}^n$ denote the input vector. Each layer performs an affine transformation followed by a nonlinearity:

$$\mathbf{z}_1 = \mathbf{W}_1\mathbf{x} + \mathbf{b}_1, \quad \mathbf{a}_1 = \sigma(\mathbf{z}_1)$$

$$\mathbf{z}_2 = \mathbf{W}_2\mathbf{a}_1 + \mathbf{b}_2, \quad \mathbf{a}_2 = \sigma(\mathbf{z}_2)$$

$$\mathbf{z}_3 = \mathbf{W}_3\mathbf{a}_2 + \mathbf{b}_3, \quad \mathbf{a}_3 = \mathbf{z}_3$$

Here, $\sigma(z) = \frac{1}{1+e^{-z}}$ is the sigmoid activation. The output layer is a linear layer, not followed by an activation.

**Backpropagation and Surrogates.** During backpropagation, we can compute $\delta$ for a neuron $i$ in the layer $l$ (going from 1 to 3 for the input to output layer) as:

$$\delta_i^l = \left( \sum_{j=1}^{n_{l+1}} W_{ij}^{(l+1)} \delta_j^{(l+1)} \right) \cdot \sigma'(z_i^l)$$

now the weight update is computed as:

$$\frac{\partial L}{\partial W_{ji}^{(l)}} = a_j^{(l-1)} \cdot \delta_i^l$$

When computing the gradient of the sigmoid, we replace the true derivative $\sigma'(z)$ with a surrogate gradient:

$$\tilde{\sigma}_k'(z) = \frac{\sigma'(kz)}{k} = \cdot \sigma(kz) \cdot (1 - \sigma(kz))$$

where $k$ controls the slope of the surrogate. Since the surrogate gradient involves both scaling the input by $k$ (to sharpen the activation) and differentiating with respect to that input, the resulting gradient scales with $k$ as well, due to the chain rule. Without compensating for this, gradient magnitudes can explode as $k$ increases. We therefore divide by $k$ to stabilize gradient flow regardless of the steepness."

**Bias in Gradient Magnitude.** The gradient of the second layer weights becomes:

$$\frac{\partial \tilde{L}}{\partial W_{ji}^{(2)}} = a_j^{(1)} \cdot \left( \sum_{h=1}^{n_3} W_{ih}^{(3)} \delta_h^{(3)} \right) \cdot \tilde{\sigma}_k'(z_i^{(2)})$$

Compared to the true gradient using $\sigma'(z)$, the surrogate gradient generally yields:

$$\tilde{\sigma}_k'(z) \geq \sigma'(z) \quad \text{for most } z$$

This introduces a positive bias in gradient magnitude, as:

$$\text{bias} = \mathbb{E}\left[ \frac{\partial \tilde{L}}{\partial W_{ji}^{(2)}} - \frac{\partial L}{\partial W_{ji}^{(2)}} \right] > 0$$

unless $k = 1$, which recovers the original derivative.

**Effect on Deep Layers.** In deeper networks, this overestimation accumulates. For example, the first-layer gradient becomes:

$$\frac{\partial \tilde{L}}{\partial W_{ji}^{(1)}} = x_j \cdot \left[ \sum_{g=1}^{n_2} W_{ig}^{(2)} \cdot \left( \sum_{h=1}^{n_3} W_{gh}^{(3)} \delta_h^{(3)} \cdot \tilde{\sigma}_k'(z_g^{(2)}) \right) \right] \cdot \tilde{\sigma}_k'(z_i^{(1)})$$

, where the sigma' are larger than their true counterparts. Compared to its counterpart using the true derivative, the surrogate gradient not only increases magnitude but distorts direction.

Besides increasing magnitude, the surrogate gradient also distorts direction. To quantify this, we compute the cosine similarity between gradients using surrogate and true derivatives:

$$\text{cosine similarity} = \frac{\nabla W \cdot \nabla \tilde{W}}{\|\nabla W\| \cdot \|\nabla \tilde{W}\|}$$

As shown in Figure 1c of the main paper, cosine similarity decays toward 0 in deeper layers when using shallow slopes (e.g., $k = 1$). This indicates that surrogate-based gradients become increasingly misaligned with the true gradient, effectively randomizing weight updates.

## 8.2 Simulator and Training Details

The design of the reward structure and termination conditions plays a critical role in shaping the learned policy. Overly strict rewards can prevent learning altogether, while rewards that are too lenient often lead to unsafe or suboptimal behavior. In practice, tuning the reward function is crucial for the success of reinforcement learning algorithms [31].

The reward function is described as:

$$r_t = C_{rs} - C_{rp}\|p_t - p_{\text{des}}\|^2 - C_{rv}\|v_t - v_{\text{des}}\|^2 - C_{rq}\|q_t - q_{\text{des}}\|^2 - C_{ra}\|a_t - C_{rab}\|^2 \quad (9)$$

where:

- $p_t \in \mathbb{R}^3$ is the current position $(x, y, z)$, $p_{\text{des}}$ is the desired hover position
- $v_t \in \mathbb{R}^3$ is the linear velocity $(v_x, v_y, v_z)$, $v_{\text{des}}$ is the desired velocity
- $q_t \in \mathbb{R}^3$ represents the orientation angles $(\theta, \phi, \psi)$, $q_{\text{des}}$ is the desired orientation
- $a_t \in \mathbb{R}^4$ is the action (motor commands)
- $C_{rs}$ is the survival bonus to discourage early termination
- $C_{rab}$ is a fixed action baseline offset

To encourage stable and progressively more precise control, a reward curriculum is applied by linearly increasing the penalties during training. The initial and final values are summarized in Table 5.

|       | $C_{rp}$ | $C_{rv}$ | $C_{ra}$ | $C_{rq}$ | $C_{rs}$ | $C_{rab}$ |
|-------|------|------|------|------|------|-------|
| Start | 1.0  | 0.01 | 0.14 | 0.25 | 1.0  | 0.667 |
| End   | 3.5  | 0.10 | 0.50 | 0.25 | 1.0  | 0.667 |

**Table 5:** Reward parameters used during training. Penalties increase linearly across epochs, updated in six steps.

**Drone Dynamics.** The simulated drone uses a simple physics model. Motor thrust is derived from RPM via a second-order polynomial:

$$T = c_0 + c_1 \cdot \text{rpm} + c_2 \cdot \text{rpm}^2$$

The motor dynamics are modeled using a first-order low-pass filter:

$$\Delta\text{rpm} = \frac{\text{rpm}_{\text{des}} - \text{rpm}_{\text{curr}}}{\tau}$$

where $\tau$ represents the motor time constant. Body-frame dynamics are integrated and then transformed to the world frame, as described by Eschmann et al.[14].

**Table 6:** Training hyperparameters for all methods.

| Parameter | BC | TD3BC | TD3BC+JSRL [Ours] | TD3 |
|---|---|---|---|---|
| **Common Parameters** | | | | |
| Hidden sizes | [256, 128] | [256, 256] | [256, 128] | [256, 128] |
| Learning rate | 1e-3 | 1e-3 | 1e-3 | 1e-3 |
| Buffer size | 1M | 1M | 2M | 2M |
| Slope start | 2 | 2 | 2 | 2 |
| Slope schedule | adaptive | adaptive | adaptive | adaptive |
| Scheduling order | 3 | 3 | 3 | 3 |
| **Method-Specific Parameters** | | | | |
| Warm-up steps | 50 | 50 | 50 | – |
| Policy noise | 0.0 | 0.0 | 0.2 | – |
| Noise clip | – | 0.5 | 0.5 | – |
| $\alpha$ (TD3BC) | – | 2.0 | 2.0 | – |
| $\tau$ (Target) | – | 0.01 | 0.01 | 0.01 |
| $\gamma$ (Discount) | – | 0.99 | 0.99 | 0.99 |
| $\lambda$ (BC coef) | – | – | 0.2 | – |
| BC decay factor | – | – | 0.99 | – |
| Training epochs | 300 | 300 | 1000 | 1000 |
| Steps per epoch | – | – | 2M | 1M |

**Training Hyperparameters.** Table 6 provides an overview of the hyperparameters used across all methods. Common parameters are listed first, followed by method-specific values.

## 8.3 Energy Consumption Analysis

Although we do not have access to neuromorphic hardware small enough to fit on the Crazyflie, the energy consumption can be estimated using the method proposed by Davies *et al.* [30], also used in [32], we see that the total energy per inference would be $9.7 \times 10^{-5}$ mJ, which is in line with the result from Wang *et al.* [32] (normalized per timestep), and also in line with real-world measurements from Paredes-Valles *et al.* [33], who measured $7 \times 10^{-3}$ mJ for a much larger and deeper vision network deployed on a neuromorphic system.

An overview of this calculation is given below.

**Table 7:** Simulation Parameters and Energy Values

| Parameter | Value |
|---|---|
| Energy per synaptic spike op $P_s$ | 23.6 (pJ) |
| Within-tile spike energy $P_w$ | 1.7 (pJ) |
| Energy per neuron update $P_u$ | 81 (pJ) |
| Layer sizes $[N_{in}, N_1, N_2, N_{out}]$ | 18, 256, 128, 4 |
| Activation Sparsity $AS$ | 0.79 |

The total energy per inference is estimated as:

$$\begin{aligned}
E &= [N_1 \cdot (P_u + N_{in} \cdot P_w)] \\
&\quad + [P_s \cdot (1 - AS) \cdot N_1 + N_2 \cdot (P_u + N_1 \cdot P_w)] \\
&\quad + [P_s \cdot (1 - AS) \cdot N_2 + N_{out} \cdot (N_2 \cdot P_w)] \\
&\approx 9.7 \times 10^{-5} \text{ mJ}
\end{aligned}$$

*Note:* The multiply operations of the input encoding are ignored. The output is a linear layer that accumulates spikes, without explicit output neurons, therefore no energy for neuron updates $P_u$ are accounted for.

## Discussion

A comparison of a conventional method (traditional controller or RNN) versus our method on the Teensy would not be fair. For realizing the energy efficiency, our method depends on neuromorphic hardware that leverages sparsity and asynchronous compute, while conventional methods would not be able to profit as much. Therefore, the hardware implementation section of our paper aims at demonstrating the feasibility of neuromorphic control, not at showing improved energy efficiency.

### 8.4 Pseudo Code: TD3BC+JSRL (Sequence-Based Training)

---

**Algorithm 1** TD3BC+JSRL for Online Spiking Policy Training

---

1: **Initialize:** Replay buffer $\mathcal{D}$, environment $\mathcal{E}$, guiding policy $\pi_{\text{ctrl}}$
2: **Initialize:** Spiking actor $\pi_\theta$, critics $Q_{\phi_1}, Q_{\phi_2}$, target networks $\pi_{\theta'}, Q_{\phi'_1}, Q_{\phi'_2}$
3: **Hyperparameters:** $\alpha$ (TD3BC weight), $\lambda_{\text{BC}}$ (BC coefficient), $\tau$ (soft update rate), $\gamma$ (discount), $\sigma_{\text{explore}}$ (exploration noise), $t_{\text{warmup}}$ (warm-up steps), $\beta$ (BC decay)
4: **for** each training iteration **do**
5:     **// Collect trajectories**
6:     **for** each rollout **do**
7:         Reset environment: $s_0 \sim \mathcal{E}$
8:         **for** timestep $t = 0$ to $T - 1$ **do**
9:             **if** $t < t_{\text{warmup}}$ **then**
10:                 $a_t \leftarrow \pi_{\text{ctrl}}(s_t)$                               ▷ Privileged guiding policy
11:                 $\pi_\theta(s_t)$                                ▷ Warm up spiking hidden states
12:             **else**
13:                 $a_t \leftarrow \pi_\theta(s_t) + \epsilon, \epsilon \sim \mathcal{N}(0, \sigma_{\text{explore}})$
14:                 $a_t \leftarrow \text{clip}(a_t, -2, 2)$
15:             **end if**
16:             Execute $a_t$, observe $(s_{t+1}, r_t, d_t)$
17:         **end for**
18:         Slice trajectory into overlapping sequences $\tau = \{(s_t, a_t, r_t, d_t, s_{t+1})\}_{t=1}^{L}$ with stride $\Delta$
19:         Add sequences $\tau$ to $\mathcal{D}$
20:     **end for**
21:     **// Network updates**
22:     **for** each gradient step **do**
23:         Sample mini-batch of sequences $\{\tau_i\}_{i=1}^{B}$ from $\mathcal{D}$
24:         **for** each sequence $\tau = \{(s_t, a_t, r_t, d_t, s_{t+1})\}_{t=1}^{L}$ **do**
25:             **Critic update:** for all timesteps $t \in [1, L]$
26:                 $y_t = r_t + \gamma(1 - d_t) \min_j Q_{\phi'_j}(s_{t+1}, \pi_{\theta'}(s_{t+1}))$
27:                 $\phi_j \leftarrow \phi_j - \nabla_{\phi_j} \|Q_{\phi_j}(s_t, a_t) - y_t\|^2$, for $j = 1, 2$
28:             $\hat{a}_t \leftarrow \pi_\theta(s_t); \quad \hat{Q}_t \leftarrow Q_{\phi_1}(s_t, \hat{a}_t)$
29:             **Actor update:** only for $t \geq t_{\text{warmup}}$
30:                 $\mathcal{L}_{\text{BC}} = \|a_t - \hat{a}_t\|^2$
31:                 $\lambda = \alpha/\text{mean}(|\hat{Q}_t|)$
32:                 $\theta \leftarrow \theta - \nabla_\theta(-\lambda \hat{Q}_t + \lambda_{\text{BC}} \mathcal{L}_{\text{BC}})$
33:         **end for**
34:         **Soft update targets:**
35:         $\theta' \leftarrow \tau\theta + (1 - \tau)\theta'$
36:         $\phi'_j \leftarrow \tau\phi_j + (1 - \tau)\phi'_j$ for $j = 1, 2$
37:     **end for**
38:     **Decay BC coefficient:** $\lambda_{\text{BC}} \leftarrow \beta\lambda_{\text{BC}}$
39:     **if** curriculum learning enabled **then**
40:         $\mathcal{E}$.update_curriculum()
41:     **end if**
42: **end for**
43: **return** trained spiking policy $\pi_\theta$

---

