# OpenReview forum: "Adaptive Surrogate Gradients for Sequential Reinforcement Learning in Spiking Neural Networks"
_NeurIPS.cc/2025/Conference — NeurIPS 2025 oral_

### Official Review · Reviewer_eoYk · 2025-06-26

**Clarity:** 3
**Significance:** 2
**Originality:** 2
**Rating:** 5
**Confidence:** 3

**Summary:**

This paper investigates the training of Spiking Neural Networks (SNNs) for continuous control tasks using reinforcement learning. The authors identify two primary challenges: understanding the optimal surrogate gradients used to train SNNs, and overcoming the difficulty of training networks in RL environments with the limited length in early training.

To address these, the paper provides a systematic analysis of how the slope of the surrogate gradient function affects gradient propagation and learning performance. Secondly, it proposes a training framework, TD3BC+JSRL, which combines several existing techniques. This method uses a privileged "guide" policy to bootstrap the learning process and generate longer initial trajectories, an off-policy RL algorithm (TD3), and a behavioral cloning term to leverage the guide's demonstrations. The authors demonstrate their approach on a simulated drone position control task and show successful transfer to a real-world Crazyflie platform.

**Questions:**

1) What is the network architecture of your work? RNN? MLP? Please provide a full discription of it.

2) The paper's motivation heavily relies on the energy efficiency of SNNs. It is encouraged to further investigate the energy efficiency on such edge situation: a flying robot, where batteries are heavy, so that energy is a much more important aspect to consider, compared to other edge computation scenarios.

3) Could you please compare your result with other SNN - RL works, to better clarify your contribution? For example, how about a E-prop trained SNN, or a Evolution trained SNN? Here's some papers that might relevant: Evolving connectivity for recurrent spiking neural networks; A solution to the learning dilemma for recurrent networks of spiking neurons.

**Ethical Concerns:**

["NO or VERY MINOR ethics concerns only"]

**Final Justification:**

The response and discussion from authors have clarified my questions, and I decided to raise my score to 5, Accept.

There are two major reasons for the raise of my rating:
A) SNN is inheriently temporal, therefore the best application for it is temporal controlling, for instance, flying control as the authors did. It  could be a possible significant application for SNNs, rather than tackling the tasks that is already accomplished by feedforward ANN architectures.
B) The authors provide a novel training technique for SNN: scheduling the surrogate gradient, and provide a thorough and detailed analysis on it. It helps the stability of training SNN, especially on scenario of the paper.

Therefore, after a revision, clarifying technical details and highlighting the significance, the paper is qualified enough to be accepted.

**Limitations:**

The primary limitation of this paper is its novelty and insufficient baseline comparisons. The proposed TD3BC+JSRL framework is a composite of well-established techniques. And the experimental evaluation is limited to variants of the TD3 algorithm.  The paper would be significantly stronger if it included direct comparisons against other relevant SNN-RL methods mentioned in the related work.

**Paper Formatting Concerns:**

The paper is carefully writen and well structured. There's not significant formatting concerns.

However, the paper contains only 8 pages. I encourage the authors to further extend the experiments in the paper, and refine in into a more infomative paper.

**Quality:**

3

**Strengths And Weaknesses:**

# Strength

1) A promising research direction. The work addresses the important and challenging problem of applying energy-efficient SNNs to real-world robotics, which is a promising research direction. Notably, flying robotics are highly energy sensitive, therefore it best suits the energy efficient SNNs.

2) Thorough Empirical Analysis: The analysis of surrogate gradient slopes (Section 4.1 and 4.2) is a valuable empirical study. It provides practical insights into how this key hyperparameter affects training stability and performance, which will be useful for other researchers in the field.

3) Valid Experimental Results: The proposed method achieves impressive performance on the drone control task, significantly outperforming the baseline methods (BC, TD3, and TD3BC) under the same conditions. The successful sim-to-real transfer is a notable achievement.

# Weakness:
The primary weakness of this paper is its limited novelty. While the empirical results are strong, the core methodological contributions are incremental and rely heavily on combining existing methods without introducing a new, fundamental concept.
The proposed TD3BC+JSRL framework is a composite of well-established techniques. Additionally, the use of surrogate gradients is the standard approach for training SNNs. The paper's analysis is an exploration of this existing technique, not the introduction of a new one.
The main contribution is the specific application and successful integration of these existing components to the problem of training recurrent SNNs. While this is a solid piece of engineering and produces good results, it does not constitute a significant scientific advance.

---

> ### Author Rebuttal · Authors · 2025-07-30
>
> We thank the reviewer for the detailed and actionable feedback. We thank you for your feedback on our sim-to-real transfer as this is a time-consuming process which usually goes undervalued.
>
> **Core Novelty and Contributions**
>
> The reviewer mentions a reasonable concern about the novelty and contributions of our work. We feel the need to further clarify this in this section and acknowledge the need to emphasize this in the camera-ready paper.
>
> A core novelty of our work is a scheduled approach to setting the slope, parameterizing this surrogate gradient. This slope scheduling method is an innovative component that brings critical robustness to SNN training, eliminating the need for extensive hyperparameter sweeps that would be prohibitive for time-consuming RL experiments. To build an understanding of the underlying effect of this parameter on the gradient propagated throughout the network, we include the surrogate gradient analysis.
>
> Our analysis reveals a fundamental trade-off: shallower surrogate gradient slopes enable more gradient updates (allowing more neurons to learn) but worsen the approximation of the true binary spiking function, an effect that compounds through network layers. This finding is supported by the theoretical framework of Gygax and Zenke in their paper: *Elucidating the theoretical underpinnings of surrogate gradient learning in spiking neural networks*, who rigorously analyzed the sign reversal which may occur when using surrogate gradients in small networks. We extend this understanding to deep networks, showing that not only sign reversal probability increases, but cosine similarity between the true and surrogate gradient decreases with a decreasing slope setting. We display how this affects RL settings more than supervised learning settings, providing crucial insights for practitioners.
>
> Finally, we find that scheduling the surrogate gradient to adapt during training allows for near-optimal performance in all test cases. Where the optimal surrogate gradient setting varies significantly, the scheduled approach succeeds in approaching the optimal surrogate gradient consistently.
>
> We emphasize a second key insight: in scenarios where the quality of the SNN controller directly impacts the length of collected sequences, such as UAV control, where poor policies lead to crashes and thus very short trajectories, the SNN struggles to converge to effective behavior. These short sequences do not provide enough temporal context to stabilize the hidden states, which is necessary for learning. This challenge is particularly acute in hard continuous control tasks of unstable systems like drone flight, where the SNN must exploit its temporal dynamics over multiple timesteps to maintain stability, distinguishing our work from simpler discrete control scenarios.
>
> To address this, we leverage a jump-starting agent. The function of this jump-starting agent differs significantly from the original publication, where its function is to increase the states seen by the trained agent. In our approach, the jump-starting agent is responsible only for bridging the warm-up period of the SNN, such that the hidden states can stabilize and that the generated sequences, which are used for training, are lengthy enough to learn from. We will more clearly emphasize this difference in the final version of the paper by introducing the privileged jump-starting actor purely as a stabilizing agent, rather than making the close comparison to the original Jump-Start RL paper. Furthermore, we will include an ablation study where we remove this jump-starting actor.
>
> **Network Architecture**
>
> The network architecture is an MLP with stateful spiking neurons (Leaky-Integrate and Fire neurons). These neurons can to some extent be interpreted as a hidden layer with a diagonal recurrency matrix, which is subsequently passed through a binary threshold function.
> All networks contain 2 hidden layers, with 256 and 128 neurons for the first and second layer respectively. Although some of this information was given in the Supplementary Materials, we see that this is still lacking important data. We will extend the network description in the Supplementary Materials and clarify the network architecture in Sec. 3.3, along with the definition of the policy.
>
> **Energy Efficiency Considerations**
>
> Through the analysis of the network using NeuroBench we aim to investigate the energy efficiency considerations for edge situations. The NeuroBench framework includes metrics which have been demonstrated to align closely with the energy consumption on real neuromorphic systems.
> For the real-world task specifically, we chose a task where energy efficiency is of prime importance: The CrazyFlie drone weighs only 27g, the batteries themselves weigh 7.1g, with flight times constrained to 3-5 minutes. Especially these platforms will benefit most from asynchronous event-based algorithms that can be deployed on tiny efficient neuromorphic hardware.
>
> **Comparison with Other SNN-RL Approaches**
>
> We acknowledge that comparing the method to other RL approaches for SNNs would be valuable. However, the motivation of this work came from the fact that existing work using RL and SNNs ignores the temporal dimension of SNNs by training on single transitions. Ignoring this temporal dimension, we found that the approaches which have been covered in literature earlier (such as DSQN) lead to large SNNs which require multiple forward passes per prediction to stabilize the hidden states. Not only did these not fit into the memory of the processor on the CrazyFlie, these networks were also not able to run at 100Hz. E-prop is designed for online learning, failing to reach the same performance as backpropagation through time, and is often considered to have limited scalability compared to RL. Evolutionary trained SNNs were initially tried, but failed to generate usable controllers. Our method was therefore the only method that successfully trains an SNN to fly the CrazyFlie using the temporal dimension.
>
> We sincerely thank you for your thorough review. Your feedback has helped us identify areas where our presentation can be clearer. We are confident that the camera-ready version will address your concerns while maintaining the paper's core contributions to practical SNN training for robotics.

---

> > ### Comment · Reviewer_eoYk · 2025-08-01
> >
> > Thank you for your response. The contribution is more clarified after the explaination. Please add the technical details of network architecture in the camera-ready version for the reproduciability of the work. However, there remains two questions that are not answered:
> >
> > A) What is the energy consumption of your network？In table 1, you provided a footprint, but you did not provide how much energy is comsumed in Joule. And how does the energy comsuption is compared to a only 7.1g battery? Adding this information could better show the significance of your work, demonstrating the application potential in real world fly controling.
> >
> > B) Existing RL and SNN works are NOT ignoring the temporal dimension of SNNs, and there exist other works that CAN be applied to controlling. SNNs are naturally temporally defined, and training through BPTT method is a standard way when applying it into temporal tasks. It is also a strong intuition to use SNNs for reinforcement learning tasks, especially controlling tasks. The paper of evolutionary SNN provide a control over robots, why it can no be applied to your case? In addition, a more recent paper: Exploring spiking neural networks for deep reinforcement learning in robotic tasks, it also provide a way to train SNN for controling tasks. In fact, the attempt to use SNN in RL is even earlier. Reservoirs and plasticity rules are also considered for the training of a controlling SNN. From my understanding, it is not a novelty for you that applies SNN into RL controling task. The contribution is that you provided a novel and more practical way for better SNN RL training process.
> >
> > Please keep in mind that these are not criticize of your work: the contribution of your work is clear and solid, provided a new training schedule over a slope parameter that especially suits the training of SNN reinforcement learning. My questions are raised for a better quality of this work.

---

> > > ### Author Response · Authors · 2025-08-02
> > >
> > > We share the reviewers view that our contributions are more clarified now. We will indeed add the technical details on the network as requested. We apologize for not clearly answering the remaining two questions.
> > >
> > > ---
> > >
> > > ### A)
> > >
> > > The energy consumption of the network on dedicated hardware is difficult to quantify. As we do not have access to a neuromorphic system small enough for the Crazyflie, we deployed the network on a regular microcontroller (Teensy 4.0). This does not exploit sparsity and binary activations as a neuromorphic processor would.
> > >
> > > Thus, we indicate energy via both accumulate (AC) and multiply-accumulate (MAC) operations.
> > > Although our SNN has a larger memory footprint, the total energy per inference is lower than the comparable ANN since ACs are at least 4 times more energy efficient than MACs (Horowitz, 2014), which require a multiply and add operation, without requiring an action history.
> > >
> > > Importantly, the MACs in our network originate from the encoding of the continuous sensor inputs to the spiking domain, in the first linear layer of the SNN. Eventually, these sensors will be replaced by neuromorphic spiking sensors, which would lead to ACs rather than MACs as well.
> > >
> > > We do also see the importance of energy numbers, and if we use a similar method to calculate the energy consumption as was used in the paper by Wang et al. we see that the total energy per inference is $9.7\times10^{-5} \text{mJ}$, which corresponds to the result from Wang et al. (normalized per timestep), and which is also in line with the real-world measurements from Paredes-Valles et al. who measured $7\times10^{-3}\text{mJ}$ for a much larger and deeper vision network.
> > >
> > > | **Parameter** | **Value** |
> > > | ----------------------- | --------------- |
> > > | Energy per synaptic spike op $P_s$  | 23.6 (pJ)|
> > > | Within-tile spike energy $P_w$  | 1.7 (pJ) |
> > > | Energy per neuron update $P_u$ | 81 (pJ)  |
> > > | # Layer sizes $[N_{in},N_1,N_2,N_{out}]$ | 17, 256, 128, 4 |
> > > | # Activation Sparsity $AS$ | 0.79 |
> > >
> > > The total energy per inference is computed as:
> > > $$
> > > E = [N_1 (P_u + N_{in} P_w)] + [P_s (1-AS) N_1 + N_2 (P_u + N_1 P_w)] + [P_s (1-AS) N_2 + N_{out} (N_2 P_w)] \approx 9.7 \times 10^{-5}~\mathrm{mJ}
> > > $$
> > > Multiply operations from input encoding are ignored. We will add the energy information to Table 1.
> > >
> > > ---
> > >
> > > ### B)
> > >
> > > You are completely correct that there exists recent work on RL for SNNs that do not ignore temporal dependencies; we apologize for the misunderstanding and will further specify.
> > >
> > > Zanetta et al. is a big addition to the field of RL for SNNs and poses one of the first implementations of RL training for SNNs that actually exploits time dependencies for control tasks. However, they also mention most of the work done in the field (Tang et al., for instance, which was used as the state-of-art reference in the mentioned paper) uses off-policy methods not exploiting the full potential of SNNs, as they ignore the temporal dimension.
> > >
> > > We add to this body of work by introducing off-policy learning that does exploit the full potential of SNN.
> > > Off-policy implementations excel on sample efficiency and data reusability. One of the big advantages is that we can pass real-world observations to the replay buffer, which can natively be integrated in our algorithm and helps sim-to-real transfer.
> > >
> > > As we showed in our paper, a network ignoring temporal dependencies (ANN without action history), cannot learn to fly. This emphasises the need for temporal dynamics as provided by Zanatta et al. or those provided by our method.
> > > Also, we want to stress that to the best of our knowledge, this is the first demonstration of an SNN policy trained using RL that was capable of stable flight in the real world.
> > > These specifics will be better explained in the camera-ready paper.
> > >
> > > In the exploratory phase, we did consider EAs as a serious option to learn a stable controller. However, our attempts converged to local minima that never approached the rewards generated by the RL methods, so we decided to focus on RL.
> > > Especially the approach proposed in the paper by Wang et al. offers promising options that require further investigation to show their applicability on highly unstable systems such as quadrotors in the real world. We will improve the discussion on other training methods and discuss the use of EA in our recommendations for future work.
> > >
> > > ---
> > >
> > > #### References
> > >
> > > * Wang, Guan, et al. "Evolving connectivity for recurrent spiking neural networks." *NeurIPS* (2023).
> > > * Horowitz, Mark. "1.1 computing's energy problem (and what we can do about it)." *ISSCC* (2014).
> > > * Paredes-Vallés, Federico, et al. "Fully neuromorphic vision and control for autonomous drone flight." *Science Robotics* (2024).
> > > * Tang, Guangzhi, et al. "Deep reinforcement learning with population-coded spiking neural network for continuous control." *CoRL* (2021).
> > > * Zanatta, Luca, et al. "Exploring spiking neural networks for deep reinforcement learning in robotic tasks." *Scientific Reports* (2024).
> > >
> > > ---

---

> > > > ### Comment · Reviewer_eoYk · 2025-08-02
> > > >
> > > > Thank you for the detailed response. The clarifications you've provided have successfully addressed my previous questions, and I appreciate the effort you've taken to do so. I'll be raising the mark accordingly.

---

### Official Review · Reviewer_ps6o · 2025-06-30

**Clarity:** 3
**Significance:** 3
**Originality:** 2
**Rating:** 4
**Confidence:** 3

**Summary:**

This paper focuses on the challenges of training Spiking Neural Networks (SNNs) for reinforcement learning (RL), particularly the non-differentiable nature of spiking neurons and the difficulty of training on sequences. The authors analyze the impact of surrogate gradient slope settings, finding that shallower slopes improve training and performance in RL. They propose a novel training approach using a privileged guiding policy to bootstrap learning, combined with an adaptive slope schedule. Applied to a drone position control task, this method achieves an average return of 400 points, outperforming prior techniques. The work advances understanding of surrogate gradients in SNNs and provides practical training methods for neuromorphic controllers, demonstrating successful real-world deployment on a Crazyflie quadrotor.

**Questions:**

1. The adaptive slope scheduler (Eq. 5) empirically improves training, but the paper lacks a theoretical explanation for why it works better than fixed slopes in some cases. Could the authors provide a convergence analysis (e.g., under what conditions the scheduler stabilizes training)? Also, could the authors provide a sensitivity analysis (e.g., how do choices of the weighting terms in Eq. 5 affect performance)?
2. The paper focuses on SNN-vs-SNN comparisons (e.g., TD3BC+JSRL vs. BC), but how does the method fare against ANN equivalents (to quantify energy efficiency vs. performance trade-offs)?

**Ethical Concerns:**

["NO or VERY MINOR ethics concerns only"]

**Final Justification:**

The authors partially solved my concerns. After reading the rebuttals and the reviews from other reviewers, I would like to keep the score.

**Limitations:**

Yes.

**Paper Formatting Concerns:**

No.

**Quality:**

3

**Strengths And Weaknesses:**

Strengths：
1. The paper presents a solid empirical evaluation, including comparisons with multiple baselines (BC, TD3BC, TD3) and ablation studies on surrogate gradient slopes. The results are well-supported by quantitative metrics (e.g., reward scores, gradient alignment analysis) and qualitative observations (e.g., real-world drone control).
2. The methodology is sound, combining theoretical insights (e.g., surrogate gradient analysis) with practical innovations (e.g., jump-start RL with a guiding policy). The use of adaptive slope scheduling is particularly well-motivated.
3. The work addresses critical challenges in training SNNs for RL, particularly in robotics (e.g., unstable early training, non-differentiability). The proposed method achieves 2.1 times performance gains over baselines, demonstrating practical impact.

Weaknesses
1. While the adaptive slope scheduler is promising, the paper does not fully explain why it outperforms fixed slopes in some cases but not others. A deeper theoretical analysis (e.g., convergence guarantees) would strengthen the claims.
2. The computational cost of training (e.g., wall-clock time vs. ANN baselines) is not discussed, which is important for neuromorphic applications.

---

> ### Author Rebuttal · Authors · 2025-07-30
>
> We thank the reviewer for their actionable feedback and insights. We thank you for your positive feedback on our empirical evaluation including multiple baselines.
>
> **Adaptive Slope Scheduler Analysis**
>
> The adaptive slope scheduler finds its strength in finding the optimal slope setting for the training setup, by incorporating knowledge about the current training performance (is the network still improving?) and test performance (is the network already at top performance?). Using these factors, it aims to find the optimal slope setting. This addresses a fundamental and intuitive trade-off we discovered: shallower surrogate gradient slopes allow more neurons to receive gradient updates, but at the cost of worse approximation of the true binary spiking function, an effect that compounds through network layers. This finding is supported by Gygax and Zenke's theoretical framework: *Elucidating the theoretical underpinnings of surrogate gradient learning in spiking neural networks*, which discusses sign reversal when using surrogate gradients. We extend this finding empirically to deep networks in RL settings, and by shifting focus from sign reversal to gradient alignment using the cosine similarity.
>
> Importantly, our slope scheduling method brings critical robustness to SNN training, eliminating the need for extensive hyperparameter sweeps that would be prohibitive for time-consuming RL experiments. This innovation is particularly valuable for hard continuous control tasks like drone flight, where the SNN must exploit its temporal dynamics over multiple timesteps to maintain stability, a challenge that distinguishes our work from simpler control scenarios.
>
> However, this slope setting lags behind slightly due to the update based on the historic trend of the training performance. When we pick the perfect slope from the beginning, we find that this can in fact outperform the adaptive scheduling, but that the adaptive scheduling is always competitive with the optimal slope. Instead of having to perform large hyperparameter sweeps, including the slope setting, the adaptive scheduler thus finds this optimal slope setting automatically, although sometimes delayed.
>
> Regarding convergence analysis: While a full theoretical proof is beyond this paper's scope, we can provide intuition for why adaptive scheduling works:
>
> - **Early Training (low reward):** The scheduler selects shallow slopes ($k$ between 1 and 10), maximizing gradient flow to all neurons despite poor alignment, enabling broad exploration
> - **Mid Training (improving reward):** $k$ increases, gradually trading exploration for exploitation
> - **Convergence (high reward):** The proportional term in the slope scheduler maintains $k$ at optimal levels, preventing degradation
>
> This mirrors established RL principles (epsilon-greedy, temperature scheduling) but operates in gradient space.
>
> Although we agree that mathematical convergence guarantees would strengthen our claims, we believe the intuition-based understanding and empirical evidence we provide already constitutes a significant contribution. Our work reveals fundamental insights about surrogate gradient behavior in deep SNNs and provides a practical solution that works robustly across different scenarios. This opens important avenues for future theoretical investigation into adaptive surrogate gradient methods.
>
> We believe the method we propose is intuitive and grounded in theoretical findings of previous work by Gygax and Zenke in *Elucidating the theoretical underpinnings of surrogate gradient learning in spiking neural networks*. They analyze the sign reversal of the gradient which occurs when the shallow surrogate gradient slope is used. We shift our focus from sign reversal to general gradient alignment, using the cosine similarity. To improve understanding of our methods, we will analyze the sensitivity of the components of our algorithm by removing the BC term, removing the jump-start period and using varying degrees of jump-starting agent levels, from novice to an expert policy.
>
> **Computational Cost Considerations**
>
> The computational cost of our approach compared to the baseline SNN is not discussed currently, but will be described more elaborately in the camera-ready version. In section 4.3 we mention that all experiments were trained on an Apple MacBook Pro with an M4Pro chip and 24GB of RAM. The SNN experiments took roughly 10 hours to train, with a single agent. The ANN solution was trained using thousands of parallel agents, using an RL library optimized in C, which allowed training in several minutes. This ANN network was then converted back to Python for further analysis. While we acknowledge this difference, the focus of the article has been on the proof of concept, rather than the optimization of the runtime of our algorithm in Python.
>
> **SNN vs ANN Comparison**
>
> The reviewer correctly points out that most of our paper consists of SNN-vs-SNN comparison, and we do not look at the performance of NNs with explicit recurrency (RNN, LSTM, GRU, ...). We have chosen to compare the SNN only to a regular MLP (the ANN in Sec. 4.4) since although the feedforward SNN has implicit recurrency from the membrane potential, there is no explicit recurrency to other neurons in the same layer. In this comparison, we focus on the trained agent, rather than the training pipeline, as ANN training allows for using many open-source optimized libraries, while the SNN equivalent requires mostly custom solutions, which are not optimized for speed.
>
> The biggest difference between the ANN and SNN is the need for an action history in ANN, while SNN can learn to fly the drone in a streaming fashion, where we do not need to keep an explicit action history, but rather just stream the state observations.
> With the ANN, it was necessary to input the action history along with the state observations to result in stable flight.
> Furthermore, the SNN leverages mostly Accumulate operations, which are significantly cheaper in hardware to perform, compared to the costly Multiply Accumulate (MAC) operations, found in the ANN. We will draw the connection between the energy consumption and the operations more clearly in the camera-ready version by referring to the NeuroBench harness.
>
> ---
>
> We sincerely thank you for your thorough review. Your feedback has helped us identify areas where our presentation can be clearer. We are confident that the camera-ready version will address your concerns while maintaining the paper's core contributions to practical SNN training for robotics.

---

> > ### Comment · Reviewer_ps6o · 2025-08-07
> > **Thankd for the reply**
> >
> > Thanks for the reply. It partially solved my concerns. I would like to keep the score.

---

> > > ### Author Response · Authors · 2025-08-08
> > >
> > > We are happy that we managed to address some of your concerns. We fully respect your remaining at your original score, but would really appreciate further explanation of which original raised questions have not been answered. We are committed to improving the camera-ready paper as much as possible, and your added feedback would be valuable to us in this effort.

---

### Official Review · Reviewer_FSHc · 2025-07-03

**Clarity:** 2
**Significance:** 2
**Originality:** 2
**Rating:** 3
**Confidence:** 4

**Summary:**

Spiking neural networks (SNN) have advantages for energy-constrained applications since it’s mostly sparse. The paper focuses on applying spiking neural networks (with surrogate gradients) in reinforcement learning to control a small drone. The authors introduce a hybrid approach between behavioral cloning (BC) and reinforcement learning (RL) where BC is applied using a guiding policy that produces good transitions at the beginning of learning where the contribution of BC fades away with time and the RL components becomes dominant. The paper validates the viability of their approach by real-world deployment of the algorithm into a small drone.

**Questions:**

- What does “The critic is not time variant and thus does not need a warm-up period” mean? The objective clearly applies the warmup and prevents any updates for both the actor and critic.
- Why is the critic ANN instead of SNN?
- The authors have in line 180: “We stack a history of observations, process multiple forward passes per observation and reset the SNN in between subsequent actions, similar to previous RL implementations for SNNs”. I’m confused because the authors argued that with SNNs we don’t need to have frame stacking. How can we reconcile both statements?

**Ethical Concerns:**

["NO or VERY MINOR ethics concerns only"]

**Final Justification:**

The authors made efforts to address my concerns. I increased the score accordingly. However, some concerns still remain.

**Limitations:**

The authors mention the limitations of their approach in the last section in the paper.

**Quality:**

2

**Strengths And Weaknesses:**

**Strengths:**

- The authors address an important problem, especially since it enables effective robot policies at small scale where energy consumption becomes a significant problem (e.g., tiny robots).
- The approach is validated by deploying on a real hardware which shows its promise.

**Weaknesses:**

- The role of BC is not clear. Why are SNNs unstable at the beginning of learning? The network should learn the representations and hidden states requires to perform better from the beginning.
- The mathematical part is not rigorous. Equation 1 has hard-coded numbers 100 and 50, where clearly those are hyperparameters of the algorithms that need to be tuned for each problem, right? Also, what is the significance of the value 100? Are all episodes constrained to have lengths of 100s?
- Unclear writing in the empirical evaluation lacks many details and it makes it hard to follow. The task should be described well, followed by the experiment setup and the results, and ending with a comprehensive discussion. I don’t think the authors follows this standard way of writing; instead the information is scattered and hard to follow.
- Many crucial components of the papers are missing:
    - The reward function definition
    - The task being addressed in the experiment is not defined.
    - Missing algorithm pseudocode. It’s not clear what algorithm steps are. The information is scattered all over the paper.
    - Standard bars in Figure 3.
    - Video of the performance of the drone moving in circles, eight-figures and squares. I checked the supplementary materials but couldn’t find any video. The only evidence is an edited photo that adds the drone in multiple places in the photo. I don’t think is a clear evidence of the behavior.
    - Comparison with recurrent ANN approaches.
    - Ablation study to understand the importance of each component in the TD3BC+JSRL.
    - Energy consumption on the drone when using TD3BC+JSRL against conventional approach.


**Minor issues:**

- The acronym JSRL is used in the paper without definition. It is referring to jump-start reinforcement learning, right?
- It’s not clear why SNNs have higher memory foot print as mentioned in line 232.
- Trajectory error in ANN with no action history cannot be infinity as reported in Table 2. It has to be a number that is measured by the authors. Please update it with the actual value.
- What is the meaning of altered drone in Table 2? How is different from the original drone?


Overall, I don't think the manuscript is ready in its current form for publication. I encourage the authors to keep working on this important problem and strengthen their paper by incorporating the feedback as needed.

---

> ### Author Rebuttal · Authors · 2025-07-30
>
> We thank the reviewer for the detailed and elaborate review. We thank your positive feedback on our hardware deployment as this is usually undervalued.
>
> **What is the role of BC?**
>
> As SNNs are notoriously difficult to train effectively, other works on SNNs for control resort to supervised learning (BC). To compare our methods to this learning paradigm, we have included behavioral cloning in our analysis.
>
> In our own algorithm we use a behavioral cloning term in the objective function to guide the SNN agent towards effective control behavior. This speeds up learning significantly. However, relying on BC alone does not allow for surpassing the performance of the sub-optimal guiding agent, as expected. So the role of BC in our algorithm is to initially achieve a decent baseline performance, while still allowing for further improvement using the TD3 term.
>
> **Why are SNNs unstable at the beginning of learning?**
>
> When initializing an SNN, the hidden states are initialized to zero. This causes issues during the first few timesteps, before the hidden states have converged to more realistic values for real flight. When training in closed-loop settings, we find that the network is not able to achieve a baseline performance where its own flights are sufficiently long to cover this stabilization period, and to enter a phase where training is performed under realistic hidden states.
>
> In drone control, where bad controllers cause the agent to crash and therefore generate very short sequences to train on, the network is unable to train effectively. In other environments such as grid-search environments, a bad controller might not be able to solve the task, but will still lead to lengthy sequences where training on these sequences delivers mostly gradient updates calculated under realistic hidden states conditions.
>
> Note, however, that during testing the SNN is not jump-started and takes control from the first timestep. The jump-starting using the guiding policy is only required for training, in the early phases of training, where the SNN itself can not yet bridge the warm-up period.
>
> **Mathematical Rigor and Surrogate Gradient Insights**
>
> We acknowledge the concerns about mathematical rigor. The theoretical foundations of surrogate gradients have been rigorously analyzed by Gygax and Zenke in *Elucidating the theoretical underpinnings of surrogate gradient learning in spiking neural networks*, which we cite. Their work provides the mathematical framework for understanding how surrogate gradients deviate from true gradients in small networks and cause sign reversal.
>
> Our contribution builds upon this theoretical foundation by:
> - Extending the analysis to deep networks: We empirically demonstrate how these deviations compound through layers, showing that shallower slopes enable more gradient updates but worsen the approximation of the true spiking function, an effect that intensifies in deeper architectures. In this analysis we also shift our focus from sign reversal to general gradient alignment, using the cosine similarity.
> - Revealing RL-specific dynamics: We show that in RL settings, the gradient noise from shallow slopes actually aids exploration, creating a unique trade-off not present in supervised learning
> - Providing a practical solution: Our adaptive slope scheduling method brings robustness and eliminates the need for extensive hyperparameter sweeps
>
> We emphasize that our finding about this trade-off can be explained intuitively: shallower surrogate gradient slopes allow gradients to flow to more neurons (enabling more updates), but at the cost of deviating further from the true binary spiking behavior. This fundamental trade-off compounds through network layers, as demonstrated in our empirical analysis and supported by Gygax' and Zenke's theoretical framework. This intuitive understanding is crucial for practitioners working with deep SNNs.
>
> Our slope scheduling method represents a key innovation that brings extra robustness to SNN training, preventing the need for exhaustive parameter sweeps that would be prohibitive for time-consuming RL experiments. This is particularly critical for hard continuous control tasks like drone flight, where the SNN must exploit its temporal dynamics over multiple timesteps, a challenge that distinguishes our work from simpler discrete control scenarios.
>
> While Gygax and Zenke provide the mathematical foundation for single neurons and small networks, focused on sign reversal, extending these proofs to deep networks in RL settings remains an open challenge. Although we agree that a mathematical proof would be valuable, we believe the intuitive understanding and empirical evidence we provide already constitutes a significant contribution and opens important avenues for future theoretical investigation.
>
> **Hyperparameters**
>
> Regarding Equation 1: It is indeed correct that 100 and 50 are hyperparameters, representing sequence length and number of warm-up steps respectively. These values were determined through preliminary experiments, balancing computational efficiency with learning effectiveness. For the camera-ready version, we will denote these correctly as hyperparameters.
>
> **Episode length and sampling strategy**
>
> Episodes are constrained to 500 timesteps, as discussed in section 3.2. We split up the episode into sequences of 100 timesteps, sampled starting at 0, 50, 100 ... 400 timesteps respectively. From a successful episode of 500 timesteps, we can thus sample 9 unique sequences.
>
> **Experimental Details**
>
> The task of the experiment is explained in section 3.2. The section also elaborates on the algorithm steps. However, we agree that this could be emphasized more clearly, and a pseudocode could improve the article's readability. The reward function has been described in the supplementary materials. It is the standard reward function from the Learning To Fly training simulator. For improved readability, we will include it in the main section of the camera-ready version.
>
> The drone was flown in the facilities of our institution. Large logos are scattered around the room, which would conflict with the anonymity of the submission. We do, however, understand the need for transparency and upon potential publication, we will release video material demonstrating the proposed method in a real-world scenario. The edited photo was meant as a trajectory visualization, not as primary evidence, we apologize for the confusion.
>
> Furthermore, we compared the SNN to conventional ANN solutions since the SNN also does not have explicit recurrence in the model architecture.
>
> An ablation study would indeed improve understanding of the components of the method and will be considered for the camera-ready version. We will analyze the sensitivity of the components of our algorithm by removing the BC term, removing the jump-start period, and using varying degrees of jump-starting agent levels, from novice to expert policies.
>
> **Minor Issues:**
> - JSRL indeed stands for Jump-Start Reinforcement Learning.
> - SNNs have a higher memory footprint as the hidden layers are larger, and the spiking neurons hold a memory state as well.
> - The trajectory error of the ANN was reported as infinity, as the ANN without action history during our testing never managed to fly the full trajectory and would always crash. We see that quantifying this as infinite is not correct, and will modify this in the camera-ready paper to reflect the fact that this network is not able to carry out the task (by replacing infinity with "-" and adding a footnote).
> - We understand the confusion regarding the altered drone, and should have described this in further detail. In this case, the regular 16mm long brushed motors of the Crazyflie are replaced by 20mm brushed motors that increase the thrust by about 40%. This way, the Crazyflie is much more stable while flying with the added weight from the Teensy 4.0 that was used to run our algorithm and greatly increased the practicality of testing. Besides, it also shows that our method is robust against a change in the dynamics in the real-world. We will discuss this in more detail in the camera-ready paper.
>
> **Critic architecture and training**
>
> The critic is a feedforward ANN, and does not encode any temporal information. The ANN critic is updated using the standard TD3 loss function which does not use a warm-up period. The critic is therefore updated on all 100 timesteps of the sampled sequence, leading to improved training efficiency.
>
> Additionally, ANNs display more stable training behavior, which we exploit by using an ANN critic instead of an SNN. Only the SNN will be deployed on the robot, and thus needs to benefit from the efficiency provided by the SNN.
>
> **Frame Stacking Clarification**
>
> We understand the confusion and see the need for further clarification.
>
> Common RL pipelines for SNNs reset the hidden states between predictions (and thus observations), therefore removing any temporal information capabilities of the SNN. However, a lot of information can be encoded in the hidden states of the spiking neurons. To leverage these, we need to stabilize them. These common RL pipelines pass a single observation several times through the network to output one single prediction. Before passing the next observation, the network is reset again. In the surrogate gradient analysis, we use the classic RL for SNN approach, and pass the network a history of observations to give it temporal context. In our own method, TD3BC+JSRL we enable training the agent using sequences, which removes the need for this frame stacking.
>
> We sincerely thank you for your thorough review. Your feedback has helped us identify areas where our presentation needs to be clearer. We are confident that the camera-ready version will address your concerns while maintaining the paper's core contributions to practical SNN training for robotics.

---

> > ### Comment · Reviewer_FSHc · 2025-08-06
> >
> > I thank the authors for their rebuttal. Most of my concerns are addressed. Thus, I will increase my score accordingly. However, some concerns remain, including missing ablation, missing video performance, and energy consumption analysis.

---

> > > ### Author Response · Authors · 2025-08-07
> > >
> > > Thank you for the reply, we are happy that most of your concerns have been addressed and we think the camera-ready paper will be more solid because of it. We do however see your remaining concerns and would like to address those even further.
> > >
> > > **Ablation Study:**
> > > We believe including an ablation of the BC term and of the jump-start period would demonstrate the effectiveness of the composed algorithm as a whole. This will be included in the camera ready version. We would also welcome further suggestions on specific ablations that could strengthen the evaluation of our method.
> > >
> > > **Video:**
> > > We appreciate the reviewer’s comment regarding the inclusion of a video. We fully agree that a flight demonstration would further strengthen the presentation of our methodology, especially given the robotics context and the importance of bridging the sim-to-real gap. While anonymization constraints limited our ability to share such material during the initial submission, we acknowledge that the inclusion of an anonymized video would have enhanced the clarity and impact of our results.
> > > At this stage, we are unfortunately unable to retroactively add a video, but, as mentioned in our previous reply, we will provide a detailed video demonstration upon publication. We will ensure that the camera-ready version includes a clear reference to the video, along with links to the code and all other necessary materials for full reproducibility.
> > >
> > > **Energy Consumption Analysis:**
> > > We fully understand the concern regarding energy consumption and like to further clarify our efforts on that aspect of the camera-ready paper.
> > > We see the importance of energy numbers for the spiking neural network, and although we do not have access to neuromorphic hardware small enough to fit on our Crazyflie, we can use a method to calculate the energy consumption. Using the method proposed in the paper by Wang et al. we see that the total energy per inference would be $9.7\times10^{-5} \text{mJ}$, which is in line with the result from Wang et al. (normalized per timestep), and which is also in line with the real-world measurements from Paredes-Valles et al. who measured $7 \times 10^{-3} \text{mJ}$ for a much larger and deeper vision network deployed on a neuromorphic system.
> > >
> > > An overview of this calculation is given below.
> > >
> > > **Table: Simulation Parameters and Energy Values**
> > >
> > > | **Parameter**                                | **Value**       |
> > > | -------------------------------------------- | --------------- |
> > > | Energy per synaptic spike op $P\_s$        | 23.6 (pJ)       |
> > > | Within-tile spike energy $P\_w$            | 1.7 (pJ)        |
> > > | Energy per neuron update $P\_u$            | 81 (pJ)         |
> > > | Layer sizes $\[N\_{in},N\_1,N\_2,N\_{out}]\$ | 17, 256, 128, 4 |
> > > | Activation Sparsity $AS$                   | 0.79            |
> > >
> > > \\begin{align*}
> > >     E
> > >     = & [N_1\\cdot (P_u + N_{in}\\cdot P_w)] + \\newline
> > >       & [P_s\\cdot (1-AS)\\cdot N_1 + N_2\\cdot (P_u + N_1\\cdot P_w)] + \\newline
> > >       & [P_s\\cdot (1-AS)\\cdot N_2 + N_{out}\\cdot (N_2\\cdot P_w)] \\newline
> > >     \\approx &9.7\\times10^{-5} \\text{mJ}
> > > \\end{align*}
> > >
> > > Note: the multiply operations of the input encoding are ignored. The output is a linear layer that accumulates spikes, without explicit output neurons, therefore no energy for neuron updates, $P\_u$, are accounted for.
> > >
> > > A comparison of a conventional method (traditional controller or RNN) versus our method on the Teensy would not be really fair. Our method depends on neuromorphic hardware that leverages sparsity and asynchronous compute, while conventional methods would not be able to profit as much. Therefore, the hardware implementation section of our paper aims at demonstrating the feasibility of neuromorphic control, not at showing improved energy efficiency.
> > >
> > > In the camera-ready paper, we will add the energy information to Table 1 and further explain this comparison in the accompanying section.
> > >
> > > We are confident that the planned improvements will improve clarity, transparency, and reproducibility of our work.
> > >
> > > **Reference:**
> > > Wang, Guan, et al. "Evolving connectivity for recurrent spiking neural networks." *Advances in Neural Information Processing Systems (NeurIPS)* (2023).

---

> > > > ### Comment · Reviewer_FSHc · 2025-08-08
> > > >
> > > > I thank the authors for their response. I think the ablation study needs to be included in the original submission (or provided in the rebuttal) so that the readers can assess the importance of each part of the algorithm. Similarly, the video needed to be part of the original submission or even added in the rebuttal phase. Regarding the anonymization argument, simple video editing could have removed distinguishable features from the video (e.g., blurring).

---

### Official Review · Reviewer_HZCM · 2025-07-18

**Clarity:** 3
**Significance:** 2
**Originality:** 2
**Rating:** 5
**Confidence:** 2

**Summary:**

The authors propose a method for training spiking neural networks via surrogate gradients in a variety of problem settings, from supervised learning to online RL. Because spiking neural networks aren’t differentiable, it necessitates the use of surrogates for the gradients being applied to the network. The authors also highlight the temporal nature of spiking neural networks aligns with the sequential decision making of RL methods. The authors present a thorough analysis of the effects of surrogate gradients between different layers of the network that inform their choice of a surrogate.

**Questions:**

- Where do the hidden states fit into the architecture of the SNN? I see a description of the SNN architecture in Sec. 3.2 but it only mentions the inputs and outputs when interacting with the CrazyFlie.

**Ethical Concerns:**

["NO or VERY MINOR ethics concerns only"]

**Final Justification:**

I believe this is a good paper that targets an interesting area in neural network training (spiking networks). It builds off theoretically justified work in smaller settings and helps motivate future theory work on spiking networks and surrogate models. I like the ideas, but wish there was more unified benchmark domains to understand the significance of the work. I'm confident the authors will address presentation issues. To me this is the biggest drawback of the current paper. However, if I saw this as a poster at NeurIPS, I would engage with the authors. The subject and approach are interesting to me, and I'd like to see more papers like this at NeurIPS overall, even with the weaknesses discussed.

**Limitations:**

yes

**Paper Formatting Concerns:**

- Line 78: wrold -> world
- Sec. 3.2 seems like it does not belong under the Methods. Perhaps it should be under a separate Experiments section?
- Eq. 1: Perhaps 100 and 50 should be hyperparameters rather than hard-coded into the function?

**Quality:**

3

**Strengths And Weaknesses:**

- The authors compare against a variety of methods that have been previously applied to spiking neural network training.

- The analysis of surrogate gradients help inform the design of the RL algorithm proposed.

- Sec. 4 could be better motivated. For example, I do not really understand the purposes of the experiments in 4.1 and 4.2. These sections focus on describing what is being done, but it is not really clear why we need to analyze the surrogate gradients.

- The method appears to be named TD3BC+JSRL, but this name is not introduced until Sec. 4.3. I think it would be better to introduce it in Sec. 3 (or earlier) and describe exactly what are the components of TD3BC+JSRL.

- I don’t have a clear understanding of how difficult these domains in the paper are because I am unfamiliar with them. This makes it difficult for me to judge the significance of the results.

---

> ### Author Rebuttal · Authors · 2025-07-30
>
> We thank the reviewer for the thoughtful and specific feedback. We thank your positive feedback on the analysis of the surrogate gradients to help inform the RL algorithm design.
>
> We acknowledge that Section 4 should be better motivated. The analysis of the surrogate gradients and the scheduling of this surrogate gradient aims at emphasizing the importance of this parameter in successfully training SNNs in both supervised and reinforcement learning settings.
>
> Conventionally, the slope setting for the surrogate gradient is not explicitly discussed or analyzed in current literature. However, we find that the setting for this gradient is a crucial parameter for training behavior and the final performance of the SNN. In RL settings, this effect is even more prevalent, as we show.
>
> Concerning mathematical rigor, the theoretical foundations of surrogate gradients have been rigorously analyzed by Gygax and Zenke (Elucidating the theoretical underpinnings of surrogate gradient learning in spiking neural networks), who provide mathematical proofs for how surrogate gradients approximate true gradients in small networks. Our work extends their theoretical framework empirically to address three critical gaps:
>
> - Extending the analysis to deep networks: We empirically demonstrate how these deviations compound through layers, showing that shallower slopes enable more gradient updates but worsen the approximation of the true spiking function—an effect that intensifies in deeper architectures.
> - Revealing RL-specific dynamics: We show that in RL settings, the gradient noise from shallow slopes actually aids exploration, creating a unique trade-off not present in supervised learning.
> - Providing a practical solution: Our adaptive slope scheduling method brings robustness and eliminates the need for extensive hyperparameter sweeps.
>
> While Gygax and Zenke provide the mathematical foundation for single neurons and small networks, extending these proofs to deep networks in RL settings remains an open challenge. We believe our empirical analysis and practical insights provide valuable contributions that complement the existing theory and can inspire future theoretical extensions to deep networks and RL contexts.
>
> We discuss that no single optimal setting for the surrogate gradient slope exists, but that this depends on task performance, learning paradigm, etc. We find that scheduling the slope of the surrogate gradient can consistently compete with the optimal slope setting across all experiments.
>
> In Section 4.1, we aim to provide the reader with a fundamental understanding of the effect of the surrogate gradient slope setting on the gradient that is propagated throughout the network. In Section 4.2, we discuss how this affects training across both supervised learning (BC) and online RL (TD3), which we believe cover the two extremes of the spectrum we consider in our article.
>
>
> For the camera-ready version, we will restructure the method section to clearly introduce the final proposed method, TD3BC+JSRL, formally. We understand the authors' concern and believe this will improve the readability of the article.
>
> We recognize the need for a more common benchmark to display the performance of our method, but believe that the chosen task of low-level quadrotor control covers several challenging aspects which our controller needs to handle. From a robotics standpoint, we believe quadrotors pose an ideal use case for neuromorphic solutions due to the high control frequency required to control these platforms and the stringent requirements on computational resources.
>
> From a learning perspective, the task is interesting as the platform is inherently unstable and very sensitive to control inputs, which makes it even challenging to hover. We focus on the lowest level control (motor-level) where related work usually focuses on a more abstracted level of control, with classic control solutions translating the higher-level controls (e.g., orientation) to motor commands. End-to-end control remains challenging with current solutions.
>
>
> Where do the hidden states fit into the architecture of the SNN?
> SNNs leverage neurons which charge membrane potential. This can be seen as a type of self-recurrency which is passed through a step function to compute the output of the neuron (and consequently release the built-up charge).
>
> ---
>
> We sincerely thank you for your thorough review. Your feedback has helped us identify areas where our presentation can be clearer. We are confident that the camera-ready version will address your concerns while maintaining the paper's core contributions to practical SNN training for robotics.

---

> > ### Comment · Reviewer_HZCM · 2025-08-04
> >
> > Thank you for the response. I am confident the authors will address my presentation concerns given their response.

---

### Comment · Area_Chair_Kt95 · 2025-08-06

Dear reviewers,

As reviewers, you are expected to stay engaged in discussion.

-  It is not OK to stay quiet.
-  It is not OK to leave discussions till the last moment.
-  If authors have resolved your (rebuttal) questions, do tell them so.
-  If authors have not resolved your (rebuttal) questions, do tell them so too.

Please note that, to facilitate discussions, Author-Reviewer discussions were extended by 48h till Aug 8, 11.59pm AoE.

Best regards,
  NeurIPS Area Chair

---

### Note · Authors · 2025-08-11

We sincerely thank all reviewers for their time, effort, and constructive feedback. We are encouraged by the positive reception and the recognition of our work’s contributions.

We thank HZCM for recognizing our analysis of surrogate gradients for RL.
We appreciate FSHc’s support in emphasizing the importance of this problem for tiny robots, and we are grateful for their compliments on our real-world deployment.
We thank ps6o for acknowledging both our theoretical contributions and the practical impact of our proposed method.
Finally, we are grateful to eoYk for their positive feedback on our empirical analysis and experimental results.

---

We've identified some areas of our paper that would benefit from modifications and have adressed these below:

We will improve general readability by introducing TD3BC+JSRL earlier in the text, restructuring the methods and experiment sections for clarity, providing a clearer motivation for the surrogate gradient analysis, and explicitly including the full network architecture (MLP with LIF neurons, layer sizes) in the methods section. We will also move the reward function definition from the supplementary material into the main text, and address smaller issues such as replacing the infinity symbol in table 2, to $N/A$ or $[-]$, explaining the altered drone setup and marking the parameters in Eq. 1 correctly as hyperparameters.

Furthermore, we will provide an ablation study that removes the BC term, removes the jump-start period, and varies the expertise of the guiding agent (novice vs. expert). This will also include additional comparisons and clarifications regarding ANN baselines and temporal dependencies.
A video displaying the flight of the drone will be referenced as a footnote. While we would have liked to include it in the rebuttal phase, this was not allowed by the NeurIPS rebuttal guidelines.

We will include an energy consumption estimate for the SNN ($\approx9.7\times10^{-5}mJ$) based on the methodology described in our responses to the reviewers. A wall-clock time for training will be included in the experiment details as well.

Finally, we will place stronger emphasis on the novelty of our surrogate gradient slope scheduling method, highlighting its role in enabling stable SNN training for hard continuous control tasks and in eliminating costly hyperparameter sweeps.

---

### Decision · Program_Chairs · 2025-09-17

**Decision:**

Accept (oral)

**Comment:**

The authors investigate the challenges of training Spiking Neural Networks (SNNs) for continuous control in reinforcement learning (RL) setups and the impact of surrogate gradients. They propose a novel training approach and apply it to a drone control task.
Strengths:
- tackles an important problem with practical relevance
- approach is validated on real hardware
- solid empirical evaluation and comparison to other methods, and ablation studies
- it shows clear performance gains
Most concerns have been addressed and clarified during the discussion period. The authors provided an energy consumption analysis in the rebuttal, having a more detailed analysis would have been beneficial. Due to the many strengths of the manuscript with only minor remaining weaknesses, I propose acceptance.